# The Effect of Low-Intensity Pulsed Ultrasound on Bone Regeneration and the Expression of Osterix and Cyclooxygenase-2 during Critical-Size Bone Defect Repair

**DOI:** 10.3390/ijms25073882

**Published:** 2024-03-30

**Authors:** Darian Volarić, Gordana Žauhar, Jie Chen, Ana Terezija Jerbić Radetić, Hrvoje Omrčen, Antonio Raič, Roko Pirović, Olga Cvijanović Peloza

**Affiliations:** 1Department of Physical Medicine and Rehabilitation, Thalassotherapia Crikvenica—Special Hospital for Medical Rehabilitation, Gajevo Šetalište 21, 51260 Crikvenica, Croatia; darian.volaric7@gmail.com; 2Doctoral School of Biomedicine and Health, Faculty of Medicine, University of Rijeka, Braće Branchetta 20/1, 51000 Rijeka, Croatia; 3Department of Medical Physics and Biophysics, Faculty of Medicine, University of Rijeka, Braće Branchetta 20/1, 51000 Rijeka, Croatia; 4Faculty of Physics, University of Rijeka, Radmile Matejčić 2, 51000 Rijeka, Croatia; 5Department of Electrical and Computer Engineering, University of Alberta, Edmonton, AB T6G 2V4, Canada; jc65@ualberta.ca; 6Academy for Engineering and Technology, Fudan University, Shanghai 200433, China; 7Department of Anatomy, Faculty of Medicine, University of Rijeka, Braće Branchetta 20/1, 51000 Rijeka, Croatia; ana.jerbic.radetic@uniri.hr (A.T.J.R.); olga.cvijanovic@medri.uniri.hr (O.C.P.); 8Department of Clinical Microbiology, Teaching Institute of Public Health of Primorsko-Goranska County, Krešimirova 52a, 51000 Rijeka, Croatia; hrvoje.omrcen@zzjzpgz.hr; 9University Integrated Undergraduate and Graduate Study Programme of Medicine, Faculty of Medicine, University of Rijeka, Braće Branchetta 20/1, 51000 Rijeka, Croatia; araic@student.uniri.hr (A.R.); rpirovic@student.uniri.hr (R.P.)

**Keywords:** low-intensity pulsed ultrasound, autologous bone, bone regeneration, critical-size bone defect, cyclooxygenase-2, osterix

## Abstract

Low-intensity pulsed ultrasound (LIPUS) is a form of ultrasound that utilizes low-intensity pulsed waves. Its effect on bones that heal by intramembranous ossification has not been sufficiently investigated. In this study, we examined LIPUS and the autologous bone, to determine their effect on the healing of the critical-size bone defect (CSBD) of the rat calvaria. The bone samples underwent histological, histomorphometric and immunohistochemical analyses. Both LIPUS and autologous bone promoted osteogenesis, leading to almost complete closure of the bone defect. On day 30, the bone volume was the highest in the autologous bone group (20.35%), followed by the LIPUS group (19.12%), and the lowest value was in the control group (5.11%). The autologous bone group exhibited the highest intensities of COX-2 (167.7 ± 1.1) and Osx (177.1 ± 0.9) expression on day 30. In the LIPUS group, the highest intensity of COX-2 expression was found on day 7 (169.7 ±1.6) and day 15 (92.7 ± 2.2), while the highest Osx expression was on day 7 (131.9 ± 0.9). In conclusion, this study suggests that LIPUS could represent a viable alternative to autologous bone grafts in repairing bone defects that are ossified by intramembranous ossification.

## 1. Introduction

It is known that the propagation of ultrasound through tissues can cause thermal and mechanical effects (micro-motion). These effects are used in physical medicine and physiotherapy to achieve clinical effects such as warming tissues, increasing cell activity and reducing pain [1,2]. The application of high-intensity therapeutic ultrasound is primarily based on its thermal effects [3], while the effectiveness of low-intensity therapeutic treatments is primarily based on non-thermal or mechanical effects such as acoustic streaming [4].

Low-intensity pulsed ultrasound (LIPUS) is a specific type of ultrasound that delivers energy to the tissue at a low intensity and outputs in pulsed wave mode [5]. The spatial average temporal average intensity (I_SATA_) of LIPUS is usually less than 100 mW/cm^2^. The currently available LIPUS devices typically emit ultrasound with an intensity of 30 mW/cm^2^ SATA (spatial average–temporal average) at a frequency of 1.5 MHz, pulsed at 1 kHz and a duty cycle of 20%. At an intensity as low as 30 mW/cm^2^, the cavitation effect can be excluded and the thermal effect with a heating of about 3 °C is negligible [6].

The use of low-intensity pulsed ultrasound to accelerate bone healing was first published by Xavier and Duarte in 1983 [7]. This was followed by clinical studies showing that the use of LIPUS accelerates the healing of fresh fractures [8,9]. The successful application of LIPUS in clinical research in the USA led to the approval of the first devices for commercial use by the US Food and Drug Administration Agency (FDA), and the registered indications for use have been the healing of fresh fractures since 1994 and the treatment of nonunions since 2000 [10,11]. The use of LIPUS has also been shown to be effective in the regeneration of soft tissues, such as tendons, ligaments and cartilage by stimulating the synthesis of fibroblasts and collagen as well as angiogenic, chondrogenic and osteogenic activity [12,13,14]. More recently, studies have investigated the possibility of using low-intensity ultrasound in dental treatment to stimulate bone formation and osseointegration of dental titanium implants [15].

Although numerous in vivo and in vitro studies have established that LIPUS stimulates the healing of bone fractures, the mechanisms are not yet fully understood. Conceivable mechanisms include direct and indirect mechanical effects such as acoustic radiation force, acoustic streaming, propagation of surface waves, fluid-flow-induced circulation and redistribution of nutrients, oxygen and signalling molecules [16,17,18,19,20].

It has also been determined that the mechanical stimulus facilitates the transformation of mesenchymal stem cells (MSC) into osteogenic cells and induces bone tissue cells to produce osteoid and its mineralization [21,22]. When LIPUS was applied in cell cultures, a significant differentiation of MSC into osteogenic cells and the development of immature preosteoblasts into mature, differentiated osteoblasts was observed, which can be attributed to the activation of gap junctions [23]. In addition, LIPUS induces MSC adhesion and proliferation, leading to the formation of clusters as future ossification centres [24,25].

Some studies suggest that a key molecule stimulated by LIPUS is the enzyme cyclooxygenase-2 (COX-2) which mediates the production of prostaglandin E2 (PGE2). The proposed mechanism is described in the manner of LIPUS producing nano-motions at the bone defect site and the nano-motion being detected by transmembrane receptors, integrins, which convert the biomechanical wave into a biochemical. The integrins then influence COX-2 which mediates PGE2 production which interacts with surrounding cells and enhances the bone regeneration process [25]. Accordingly, LIPUS should represent a valuable non-invasive bone healing alternative in view of the fact that COX-2 plays a key role in both endochondral and intramembranous bone formation during skeletal repair [26].

With regard to methods that can stimulate bone healing, autologous bone grafts (AB) are considered the “gold standard” as they exhibit osteoconductive, osteoinductive and osteogenic properties. The osteoconductive properties manifest themselves in the form of a scaffold in which the AB binds osteocytes and osteoblasts to itself and thus promotes the healing of bone defects from the centre to the edges. The osteoconductive properties depend on the three-dimensional structure of the graft and determine the speed of osteointegration. The AB graft is also osteoinductive as it contains growth factors, matrix proteins and signalling molecules that promote the bone healing process. The osteogenic properties are demonstrated by the fact that they activate osteogenic precursor cells and osteoblasts, which will be involved in the formation of new bone. The most common donor site for AB in humans is the iliac crest, which is mainly used for larger bone defects. Alternative donor sites are the proximal tibia, the calcaneus, the greater trochanter of the femur and the distal radius. The most common complication of AB graft harvesting is pain at the donor site. Other complications such as nerve injury, haematoma or infection can occur but are the exception rather than the rule [27]. The main disadvantage of the AB graft is the fact that it degrades rapidly. This is demonstrated by the fact that almost 40% of the bone volume is lost during the bone healing and remodelling process. Due to this fact, it is not uncommon for AB transplants to be combined with xenografts (bone substitute material obtained from animal species such as cattle, pigs, and horses), as these offer advantages in terms of their mechanical properties and resilience to resorption so that bone volume stability is achieved [28].

A critical-size bone defect (CSBD) is the smallest bone defect which will not heal spontaneously during the life span of an animal. When referring to the rat calvarial defect, 8 mm is mainly accepted to be of critical size. Nevertheless, smaller defects have been examined in models with two defects per animal, permitting fewer animals to be necessary for a given study [29]. Considering that the healing of the calvary bone is identical to the type of ossification during embryonic development and since it ossifies by means of intramembranous ossification, it is evident that the bone defect in the calvary of the animal represents the best model for intramembranous bone healing [30,31]. During healing, the bone of the rat goes through a few phases: (1) On days 0–1, the blood clot begins to form, platelets secrete TGF and PDGF, and inflammatory cells produce pro-inflammatory cytokines: interleukin 1 (IL-1), interleukin 6 (IL-6) and tumour necrosis factor-alpha (TNF-α). The COX-2 enzyme is upregulated during the inflammatory phase. Its role in the production of prostaglandins contributes to the initiation of the inflammatory response and the recruitment of inflammatory cells to the bone defect site. MSC create various bone morphogenetic proteins (BMPs) which have a role in proliferation, mitogenesis, angiogenesis, and chemotaxis. BMPs initiate the differentiation of osteoprogenitor cells (OPC) into osteocytes. (2) During days 2–5, the increase in MSCs and osteoblast differentiation takes place at intramembranous ossification locations. Woven bone is formed by osteoblast differentiation from the cortical bone and the inner periosteum layer. Subsequently, a reduction in inflammatory cytokine levels and an increase in Runt-related transcription factor 2 (Runx-2) occurs. Runx-2 represents one of the key factors of osteoblast differentiation. Osterix (Osx), another transcription factor acts downstream of Runx2 and is responsible for the transition of MSC to preosteoblasts. The expression of Osx is highly specific to osteoblasts and preosteoblasts, making it a useful marker for studying bone formation. Osx has been recognized as the most highly expressed transcription factor during the final stages of osteoblast differentiation within the newly formed bone and it stimulates the formation of osteocalcin (OCN), collagen and osteopontin (OPN) thus supporting bone remodelling [32,33,34]. (3) During days 7–10, the peak of proliferation of osteoblasts and OCN expression occurs. On day 14, proliferation is decreased, while osteoblastic osteoid production resumes, which mineralizes the binding callus and produces woven bone. Neoangiogenesis is induced, which is followed by the expression of vascular endothelial growth factor (VEGF). From days 14 to 21, osteogenesis is the most active and there is a second elevation of pro-inflammatory cytokines (IL-1, IL-6, TNF-α) which corresponds with bone remodelling. During days 21 to 35, woven bone is remodelled into mature lamellar bone which is indicated by a reduction in TGF-β expression and elevation of sclerostin expression [35,36,37].

Available studies regarding LIPUS application on rat calvarial defects are not plentiful, and a small number of published studies report a positive effect of LIPUS on calvarial defect healing [38,39,40,41]. Nevertheless, no studies have yet been conducted which have examined the dynamics of CSBD healing after LIPUS application nor have any studies compared this defect after AB graft implementation. Correspondingly, no studies which have performed the immunohistochemical analysis of Osx and COX-2 on the critical-size bone defect treated with LIPUS, have been executed.

Considering the mentioned facts, we hypothesized that both LIPUS and AB would stimulate osteogenesis and that their application would lead to the complete closure of the critical-size bone defect of the rat calvary. Furthermore, we expected that the bone volume values would depend on healing dynamics and would differ between LIPUS and AB experimental groups. Additionally, we anticipated that the immunohistochemical expression of Osx and COX-2 would also depend on the healing dynamics and that their intensity would differ between the two experimental groups.

Therefore, our aims were to utilize LIPUS and AB and examine their effect on CSBD healing, not to mention, compare them to the spontaneous control group healing. Accordingly, the aim was to quantify the bone volume values in the LIPUS group compared to the AB and control group and furthermore to analyse and quantify the intensity of the immunohistochemical expression of COX-2 and osteogenic factor osterix (Osx).

## 2. Results

### 2.1. Histological Analyses

On day 7 the bone defect was bridged with fibrous tissue in all groups. The largest quantity of fibrous tissue containing clusters of inflammatory cells was observed in the LIPUS group. In addition, new bone formation was observed on the edges of the bone defect. Analogous findings were found in the control group, whereas in the AB group, the fibrous response was diminished encompassing AB particles (Figure 1a,d,g).

On day 15 the LIPUS group was characterized by newly formed bone trabeculae which bridged the centre of the defect, in addition to marginal bone formation with osteoblasts on the bone surface (Figure 1b). A single newly formed bone trabeculae in the AB group bridged the defect from edge to centre, while some graft particles were resorbed, others remained (Figure 1e). The control group was still dominated by fibrous tissue, with minimal peripheral quantities of newly formed bone on the defect edges (Figure 1h).

On day 30 LIPUS led to the closure of the defect, which was almost completely filled with woven bone, except in the central part, where abundant fibrous tissue could be seen. Lamellar bone was present on the edges of the defect (Figure 1c). When analysing the AB group, newly formed bone was found in the central part of the defect and in the lower part, where bone integration can be seen. In the upper part of the defect, there was plentiful fibrous tissue which separated newly formed bone from lamellar bone (Figure 1f). In the control group, newly formed bone was present at the defect edges, whereas the central portion was still bridged by fibrous tissue enveloping clusters of pluripotent mesenchymal stem cells in which bone deposits began to appear (Figure 1i).

### 2.2. Histomorphometric Analysis

The values of the bone histomorphometric parameter BV/TV at specific time points are shown in Table 1. The results presented show statistically significant differences between the groups.

There is also a significant difference according to specific time points. The values of newly formed bone showed statistically significant differences between day 30 and both other time points, day 15 and 7 when referring to all groups (*p* < 0.001). This statistical significance is most evident in the LIPUS group where the *p* value is the smallest (*p* = 0.000001). In addition, there was a statistically significant difference between day 30 and two other time points, day 15 and day 7 (*p* < 0.001) in all groups, LIPUS, AB group and control.

Figure 2 shows how the BV/TV (%) values progress over time, with reference to specific time points (7th, 15th, 30th day) taking all groups into account.

### 2.3. Immunohistochemical Analyses

The immunohistochemical results presented in Table 2 show that there are statistically significant differences in the intensity of COX-2 and Osx expression between the control group and the LIPUS and AB groups for each time point (*p* < 0.001). In addition, there was a statistically significant difference in all groups between day 7 and two other time points, day 15 and day 30 (*p* < 0.001). The immunohistochemical analyses of COX-2 and Osx regarding rat calvarial bone defects are shown in Figure 3 and Figure 4.

## 3. Discussion

The bones of the human body that ossify by means of intramembranous ossification include the majority of the craniofacial bones as well as the clavicle. Due to the fact that most of these bones safeguard vital structures, it is not unforeseen that if these bones succumb to injury, one should attempt to facilitate and improve their regeneration. In spite of the fact that AB is regarded to be the gold standard among bone graft substitutes, reported possible complications associated with its use shifted the focus to the development of alternatives [42].

CSBD studies have been established to be very favourable for examining different influences as well as types of biomaterials and they provide important information about osteoconductive and osteoinductive properties of biomaterials. Due to the fact that the calvary bone ossifies by intramembranous ossification, this makes the CSBD of the rat calvary the foremost model to study intramembranous ossification [43,44]. Since LIPUS has already proven to be a valuable non-invasive healing accelerator for long bone defects, which regenerate by endochondral ossification, we resolved to apply it to the CSBD of the rat calvary and compare it to the current gold standard in transplantation surgery—the autologous bone graft.

Taking this into consideration, we examined rat calvary CSBD healing histologically and immunohistochemically after applying LIPUS and the AB graft. To date, a few studies have been published in which in vivo rat calvarial bone defect healing has been well documented after LIPUS application. However, there is a deficiency regarding scientifically established data comparing LIPUS efficiency to bone defect healing using AB grafts as well as analysing immunohistological parameters such as Osx and COX-2 [38,39,40,41]. Jung et al. applied LIPUS of 1 W/cm^2^ on 4-mm calvarial bone rat defects twice a week for 20 min for 8 weeks. Their result confirmed the beneficial effects of LIPUS on new bone formation, with approximately 20% higher bone values in the experimental group as opposed to control [38]. Lavandier et al. utilized LIPUS of 100 and 300 mW/cm^2^ on 3-mm calvarial bone rat defects 5 min per day, five consecutive days per week, for two weeks after the creation of the bone defect. Imaging was performed on day 3, day 30 and day 60 after surgery. The results of this study showed a significant ultrasound effect with 300 mW/cm^2^ and not with 100 mW/cm^2^. The mean 60-day bone reconstruction ratio of the 300 mW/cm^2^ group was about 20% of volume [39]. Hasuike et al. applied 30 mW/cm^2^ LIPUS on noncritical-sized 2.7-mm rat calvarial defects for 20 min daily for a total of 28 days. At 4 weeks, a significant difference in the reossification ratio was observed (18.1% in the LIPUS group vs. 9.8% in the control) [40]. Imafuji et al. performed 4 circular defects of 2.7-mm in diameter on the rat calvary and divided the experimental animals into two main groups—LIPUS and non-sonicated group. The 30 mW/cm^2^ LIPUS applied group had three subgroups; a control group with only ultrasound applied, one group with absorbable collagen sponges (ACS) and one group with sponges combined with bone morphogenetic protein—9 (rhBMP-9/ACS). The non-sonicated group had equivalent subgroups without LIPUS stimuli. The sonication was daily and lasted for 4 weeks post-surgery. LIPUS-applied groups exhibited developed ossification compared with LIPUS-non-applied groups. Bone volume measurement revealed higher values in LIPUS-applied groups compared with LIPUS-non-applied groups [41]. In all these studies the values of newly formed bones were quantified using computed tomography.

Our study confirmed the beneficial effects of LIPUS and AB on CSBD healing in terms of almost complete bone defect regeneration, as well as differences in intensity of Osx and COX-2 expression between the analysed groups.

According to our results, bone volume varied by groups and by time points. The trend was for bone volume to increase by days, reaching the highest values by day 30. The highest values on day 30 were observed in the AB group (20.35%), followed by LIPUS (19.12%), and the lowest value in the control (5.11%). On day 30 a significantly higher bone volume was found in both the LIPUS, and AB groups compared to the control group. In addition, LIPUS showed significantly higher BV/TV on day 7 than the control group (Table 1, Figure 2).

Throughout the early repair stages, the defect area of the LIPUS group was bridged over by thickened fibrous tissue and filled by aggregates of inflammatory cells (Figure 1a). This thick bridge of fibrous tissue provided a solid model from which bone will form in the intermediate and late stages of repair.

On day 15 of the defect repair, a similar percentage of the bone volume was achieved in the LIPUS and AB groups (Table 1). However, the pattern of bone formation was different. In the AB group, the newly formed bone trabecula extended from the edge of the defect towards the AB fragment confirming that it has acted as a scaffold (Figure 1e). In the LIPUS group, bone trabeculae formed within the fibrous tissue, presumably due to the effect of the mechanical stimuli, as no bone trabeculae were formed in most samples of the control group (Figure 1b,h). During the final stage of bone defect repair, the newly formed bone had almost completely filled the defect area in the LIPUS and AB groups, while in the control group, only small deposits of bone tissue were observed within the mesenchymal cell clusters (Figure 1c,f,i).

Taking these facts into account we can conclude that both experimental agents, LIPUS and AB significantly led to the regeneration of the rat calvary CSBD compared to the spontaneous defect healing of the control group.

The immunohistochemical results showed that on day 7, a significantly higher intensity of Osx and COX-2 expression was detected in the LIPUS group compared to the AB and control groups, respectively (Table 2). The immunopositivity of COX-2 was mainly seen in inflammatory cells, suggesting that low-intensity pulsed ultrasound stimulated inflammation, which is a key event that initiates bone repair (Figure 3a). Also, Osx expression in the pluripotent mesenchymal stem cells reflects their proliferation, which preceded differentiation to osteogenic cells (Figure 4a) [43]. On day 7, the control group showed higher expression of COX-2 and OSX compared to the AB group, possibly due to a more intense inflammatory response (Table 2).

Throughout day 15, significantly higher COX-2 intensity values were found in the LIPUS group than in the other two groups (Table 2). COX-2 was immunolocalized in the osteoprogenitor cells adjacent to newly formed bone trabeculae and in the immature osteocytes in the bone lacunae (Figure 3b). It has been well documented that bone formation is stimulated by mechanical loading, yet only a few studies have reported the fact that it is regulated by COX-2 expression [45,46]. In a study by Zhang et al., it was shown that inflammation-induced intramembranous ossification was reduced in COX 2^−/−^ mice. It has also been reported that COX-2 is required for the expression of the BMP-regulated cbfa and Osx genes, which are critical for bone defect regeneration [26]. It has been demonstrated that LIPUS through mechanical stimuli and enhanced COX-2 expression also upregulates even the production of prostaglandin E2 in osteoblasts which further accelerates fracture repair [47]. Our results confirmed that the mechanical stimulus induced by low-intensity pulsed ultrasound stimulated bone formation which was mediated by COX-2.

During the intermediate defect repair stage, Osx appeared to be significantly higher in the control group compared to the two experimental groups (Table 2). Immunopositivity was observed in the osteoprogenitor cells adjacent to islets of the newly formed bone (Figure 4h). Such a result suggests Osx mediated transition of the mesenchymal stem cells into osteogenic cells. It also indicates the newly formed bone trabeculae attempting to bridge over the defect from the centre to the edge. The latter was observed in two bone samples, while three bone samples failed to form new bone in the centre of the defect.

On day 30, Osx and COX-2 were most strongly expressed in the AB group, which significantly differed compared to the LIPUS and control group (Table 2). Expression was observed in the mesenchymal cell proliferation sites and preosteoblasts of the narrow fracture sites that have not yet been ossified, as well as in newly formed bone osteocytes (Figure 3f and Figure 4f). This result indicates that both examined proteins are expressed in the osteogenic cells and pluripotent mesenchymal cells, suggesting that the capacity for bone regeneration is still present in the late stages of the repair.

Considering these results, we can determine that mechanical stimulus in the LIPUS group induced the highest Osx and COX-2 expression on day 7 day as well as the highest COX-2 expression on day 15 of defect repair. This has been reflected in the proliferation of pluripotent mesenchymal stem cells and osteogenic cell formation, which has contributed to superior bone regeneration and almost complete defect closure. COX-2 expression decreased towards later phases, while newly formed bone volume increased, suggesting that as the defect stabilises, less of the COX-2 is involved in its repair. On day 30, the expression of Osx and COX-2 was the highest in the AB group, which correlated with the highest potential of new bone formation, which, however, failed to completely integrate with the lamellar bone. The result of the significantly higher expression of Osx on day 15 in the control group was most likely induced by an unstable, fibrous, self-stimulating membrane which eventually failed to close the defect.

## 4. Materials and Methods

### 4.1. Animals and Experimental Design

The experiment was performed on 2.5-month-old male Wistar rats, with the total number of N = 45. The animals were randomly distributed into 3 groups, 15 animals per group, which were sacrificed on the 7th, 15th and 30th day of the experiment (Table 3). The critical-size bone defect was performed on all animals and covered with a collagen membrane (Mucoderm^®^, acellular dermal collagen matrix, Botiss Biomaterials GmbH, Berlin, Germany) to mimic the periosteum. In the first group, only LIPUS was applied, in the second group only the AB transplant was applied, and in the third group was the control where spontaneous critical-size bone defect healing was observed. This research was approved by the Ethical Committee for Biomedical Research of the Faculty of Medicine at the University of Rijeka as well as the Ministry of Agriculture (EP 302/2021). All animals were treated in accordance with the Guidelines of Care and Use of Animals for Experimental Procedures.

### 4.2. Experimental Equipment and Treatment

We used the LIPUS device SonaCell (IntelligentNano Inc., Edmonton, AB, Canada) to generate low-intensity pulsed ultrasound waves with the following parameters: ultrasound frequency = 1.5 MHz, pulse repetition rate = 1 kHz and pulse duty cycle = 20%. The intensity on the transducer surface was 30 mW/cm^2^. The actual output power of the transducer was measured by an ultrasonic power meter before use. The LIPUS device has a single-element circular transducer with a diameter of 24 mm. LIPUS insonation was performed after placing the experimental animal in a box that restricted excessive movements and fixed the rat head (Figure 5). The ultrasound gel was applied directly on the heads of the rats in the LIPUS group prior to ultrasound treatment. The LIPUS rats were exposed to an application of ultrasound for 20 min, three times a week (Monday, Wednesday, and Friday). The application of LIPUS started on day 1 and continued to day 30, depending on respective time points.

### 4.3. Surgical Protocol of Performing CSBD

In the frontoparietal complex, an intracranial defect was punctured using a trephine of an outer diameter of 8 mm at 1500 rpm (Figure 6). A detailed description of the surgical procedure for performing the CSBD can be found in Jerbić Radetić et al. [43].

### 4.4. Bone Tissue Processing

Upon animal sacrifice bone tissue of the rat calvary was gathered. Tissue samples were kept in a 4% paraformaldehyde solution and stored in a cold storage box at 4 °C until moving. Afterwards, the tissue samples were submerged in a 70% alcohol solution until histological staining.

### 4.5. Histological Staining

Rat calvaria bone samples were subjected to decalcination in Osteofast 2 (Biognost, Zagreb, Croatia) for two days. After the decalcination process, the bone samples were embedded in paraffin blocks after which they were cut using a microtome (Leica RM 2155-Rotary Microtome, Leica instruments, Ballerup, Germany) to reach the centre of the defect. All specimens were cut through the sections of the parietal bone, with the crista temporalis and m. temporalis observed on the lateral margins of the convex surface of each section serving as orientation points. On the concave surface of some sections, the superior sagittal sinus was observed. Specimens were cut into 3–5 μm thick tissue sections and stained with hematoxylin and eosin (HE). Region of interest (ROI) is analogous to the margins of the bone defect and the zone within defect margins.

### 4.6. Histomorphometric Analysis

On tissue sections stained with histological stain bone histomorphometry was executed. Operating a light microscope Olympus BHA (Olympus Corporation, Tokyo, Japan) mounted with a digital Sony camera (Sony Corporation, Tokyo, Japan) microphotographs of tissue sections were performed. Microphotographs of tissue sections were analysed using a semiautomatic image analysis system that employed ISSA software v2.5 (VAMS, Zagreb, Croatia). Bone Volume (BV) to Tissue Volume (TV) ratio (BV/TV, %) was computed using the standardized method of stereometric measurements proposed by Parfitt A.M. [48].

### 4.7. Immunohistochemical Analyses and the Intensity of the Immunohistochemical Expression

Xylene deparaffinization and ethanol dehydration were performed on 3–5 μm thick tissue sections. Following deparaffinization and dehydration endogenous peroxidase activity was blocked using 0.3% H_2_O_2_ in methanol and after that incubated in citrate buffer for 10 min (T = 60 °C) to identify antigen. Afterwards, rabbit polyclonal anti-Sp7/Osterix antibody (ab 22552 Abcam, Abcam, Cambridge, UK) and rabbit polyclonal anti-COX-2 antibody (PA5-27238, Invitrogen, Waltham, MA, USA) were used for overnight incubation with tissue samples. After antibody incubation, the specimens were then washed and incubated with a secondary biotinylated antibody for 45 min. For visualization, peroxidase-conjugated streptavidin (LSAB+ Kit, DakoCytomation, Glostrup, Denmark) and 3,30-diaminobenzidine (DAB, DakoCytomation, Glostrup, Denmark) were applied. Hematoxylin was used to contrast the nuclei. After applying a resin (Biomount, Biognost, Zagreb, Croatia), the slides were microscoped with an Olympus BHA light microscope (Olympus Corporation, Tokyo, Japan) mounted with a digital Sony camera (Sony Corporation, Tokyo, Japan). The intensity of immunohistochemical staining was quantitatively analysed using a computer program ImageJ v1.53 (Wayne Rasband, National Institute of Health, Bethesda, MD, USA).

### 4.8. Statistical Analyses

The statistical analysis was performed using the computer program Statistica 14.0.1.25 (TIBCO Software Inc., Palo Alto, CA, USA). The normality of the distribution was tested with the Kolmogorov–Smirnov test, which showed that the data were normally distributed. Data are expressed as means ± standard deviation (SD). Differences between groups were tested using the one-way analysis of variance (ANOVA) test, including the post-hoc Scheffe test. All data obeyed the homogeneity of variance. The statistical significance level was set at α = 0.05 (*p* < 0.05 being considered statistically significant).

## 5. Conclusions

Both low-intensity pulsed ultrasound and autologous bone stimulated osteogenesis, and their application resulted in almost complete closure of the critical-size bone defect of the rat calvary. Regardless of the fact that autologous bone contributed to the largest quantity of new bone, low-intensity pulsed ultrasound displayed envious recently developed bone volume with increased immunohistological expression of both COX-2 and Osx. Therefore, this study demonstrated the possibility of low-intensity pulsed ultrasound being a viable alternative to autologous bone grafts in the repair of bone defects concerning the craniofacial region and clavicles, which ossify by intramembranous ossification, exploring new clinical possibilities in the form of non-invasive ultrasound treatment for these types of bone defects. Further studies are needed for better understanding and efficiency optimization regarding LIPUS bone repair stimulation.

## Figures and Tables

**Figure 1 ijms-25-03882-f001:**
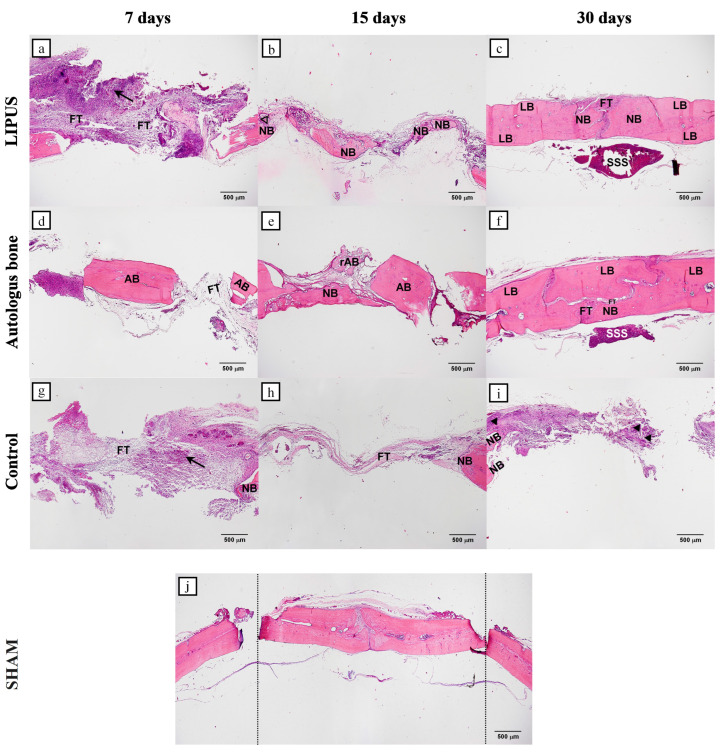
Representative microphotographs of the coronal sections of rat calvarial bone defects for all groups: LIPUS (**a**–**c**); Autologous bone (**d**–**f**) and Control (**g**–**i**). For each group, three representative time points were chosen: 7 days (**a**,**d**,**g**), 15 days (**b**,**e**,**h**), and 30 days (**c**,**f**,**i**). SHAM image (**j**) shows the bone defect in full length. In tissue section, new bone (NB), lamellar bone (LB), autologous bone (AB), resorbed autologous bone (rAB), fibrous tissue (FT), inflammatory cell clusters (black arrows), osteoblasts (no fill triangles), clusters of mesenchymal stem cells (black fill triangles), superior sagittal sinus (SSS), Region of Interest—ROI (space between dotted lines) were marked (HE staining, magnification 40×, scale bar = 500 µm).

**Figure 2 ijms-25-03882-f002:**
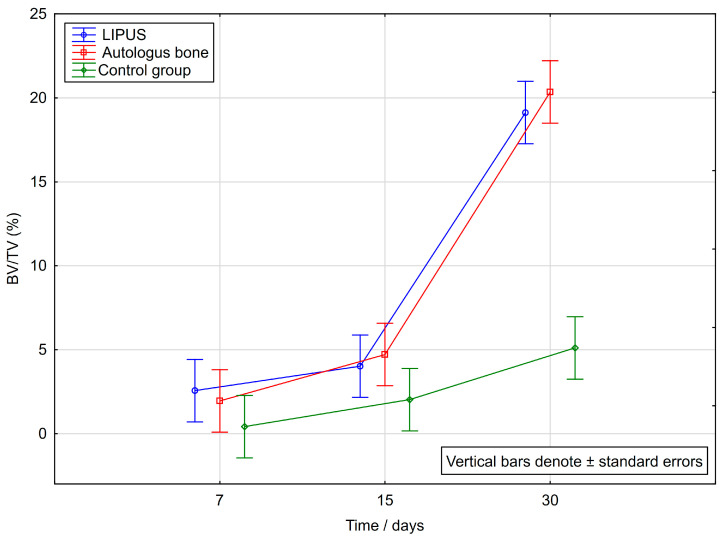
Variations in Bone Volume (BV/TV, %) by time in LIPUS, Autologous bone and Control group.

**Figure 3 ijms-25-03882-f003:**
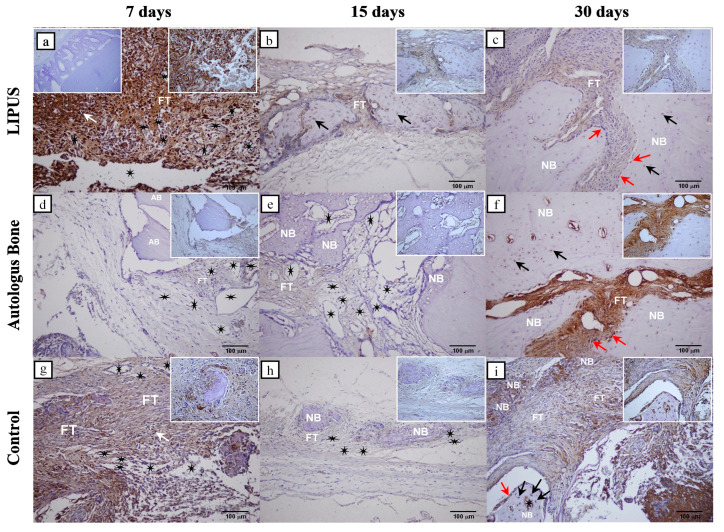
Cyclooxygenase-2 (COX-2) immunohistochemical staining of the coronal sections of rat calvarial bone defects for all groups: LIPUS (**a**–**c**); Autologous bone (**d**–**f**) and Control (**g**–**i**). For each group, three representative time points were chosen to show the temporal and spatial localization of COX-2: 7 days (**a**,**d**,**g**), 15 days (**b**,**e**,**h**), and 30 days (**c**,**f**,**i**). The negative control is shown in the left upper corner of the (**a**). Regional magnifications (400×) are shown in the right upper corner of each microphotograph. New bone (NB), autologous bone (AB), fibrous tissue (FT), inflammatory cells (white arrows), osteocytes (black arrows), osteoblasts (red arrows), blood vessels (black stars) were marked (magnification 200×, scale bar = 100 µm).

**Figure 4 ijms-25-03882-f004:**
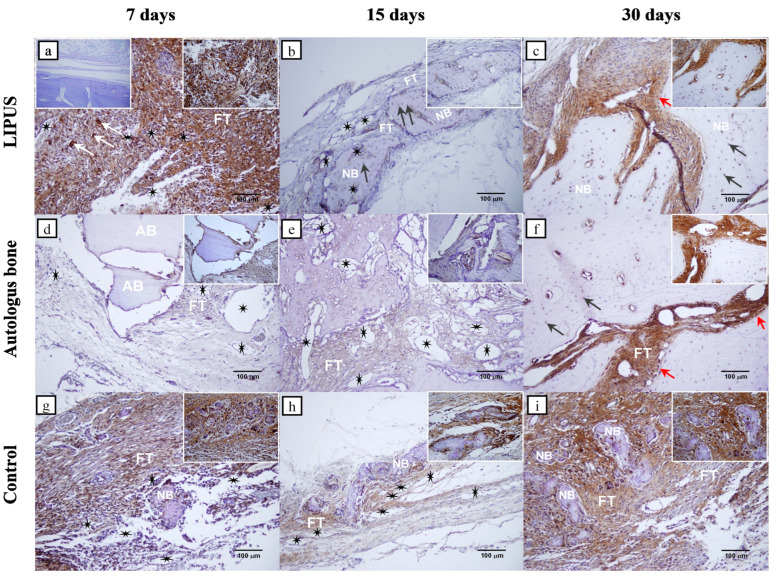
Osterix (Osx) immunohistochemical staining of the coronal sections of rat calvarial bone defects for all groups: LIPUS (**a**–**c**); Autologous bone (**d**–**f**) and Control (**g**–**i**). For each group, three representative time points were chosen to show temporal and spatial localization of Osx: 7 days (**a**,**d**,**g**), 15 days (**b**,**e**,**h**), and 30 days (**c**,**f**,**i**). The negative control is shown in the left upper corner of the (**a**). Regional magnifications (400×) are shown in the right upper corner of each microphotograph. New bone (NB), Autologous bone (AB), fibrous tissue (FT), inflammatory cells (white arrows), osteocytes (black arrows), osteoblasts (red arrows), blood vessels (black stars) were marked (magnification 200×, scale bar = 100 µm).

**Figure 5 ijms-25-03882-f005:**
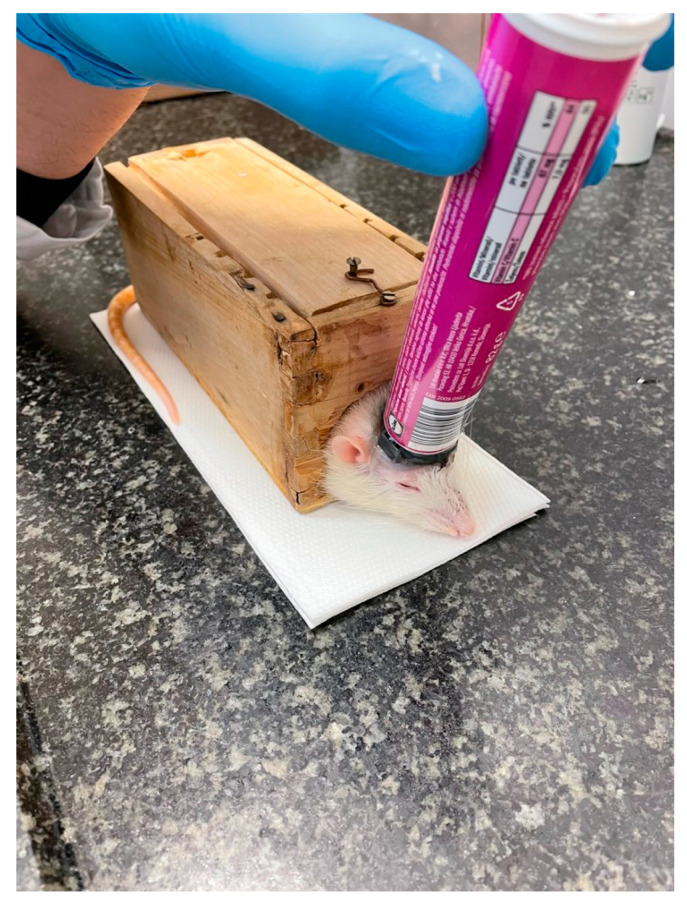
Experimental set-up and position of rats during LIPUS insonation.

**Figure 6 ijms-25-03882-f006:**
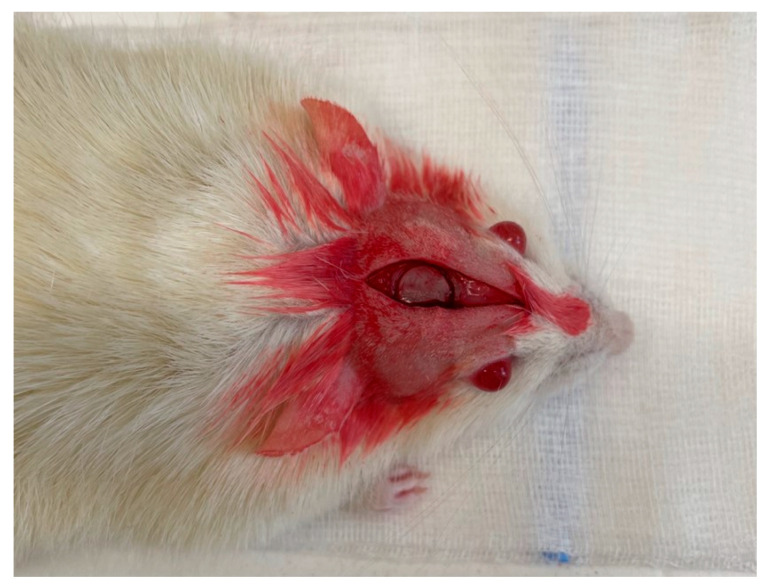
Critical-size bone defect (CSBD) of the rat calvary.

**Table 1 ijms-25-03882-t001:** Histomorphometric results of the BV/TV parameter (%) (Mean Value ± SD).

Day	LIPUS Group	Autologous Bone Group	Control Group	Overall *p*	Pairwise Differences *
7	2.56 ± 1.87	1.95 ± 1.32	0.42 ± 0.24	0.007	cl
15	4.02 ± 2.09	4.71 ± 4.65	2.03 ± 1.56	0.180	/
30	19.12 ± 8.53	20.35 ± 12.64	5.11 ± 3.71	0.002	cl, ca

* cl—significant difference between control group and LIPUS group; ca—significant difference between control group and autologous bone group.

**Table 2 ijms-25-03882-t002:** Immunohistochemical intensity score results. (Mean Value ± SD).

	Day	LIPUS Group	Autologous Bone Group	Control Group	Overall *p*-Value *	Pairwise Differences **
COX-2	7	169.7 ± 1.6	55.5 ± 1.1	131.1 ± 1.6	<0.001	cl, ca, al
	15	92.7 ± 2.2	72.5 ± 2.2	64.3 ± 1.7	<0.001	cl, ca, al
	30	81.4 ± 1.8	165.7 ± 1.1	111.0 ± 1.5	<0.001	cl, ca, al
OSX	7	131.9 ± 0.9	23.5 ± 0.9	102.2 ± 1.2	<0.001	cl, ca, al
	15	42.9 ± 0.9	64.7 ± 0.7	81.5 ± 1.8	<0.001	cl, ca, al
	30	128.9 ± 1.2	177.1 ± 0.9	130.9 ± 0.5	<0.001	cl, ca, al

* Analysis of Variance. ** cl—significant difference between control group and LIPUS group; ca—significant difference between control group and autologous bone group; al—significant difference between autologous bone group and LIPUS.

**Table 3 ijms-25-03882-t003:** Experimental design.

Group Number	Group	Number of Animals (N)	Time Points	Total
1.	LIPUS	5	3 (7, 15, 30 days)	15
2.	Autologous bone	5	3 (7, 15, 30 days)	15
3.	Control	5	3 (7, 15, 30 days)	15

## Data Availability

The data presented in this article are available on request from the corresponding author.

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
