# Peer review of "The Effect of Low-Intensity Pulsed Ultrasound on Bone Regeneration and the Expression of Osterix and Cyclooxygenase-2 during Critical-Size Bone Defect Repair"

_ijms, 2024, doi:10.3390/ijms25073882_

Round 1
Reviewer 1 Report
Comments and Suggestions for Authors
In this manuscript, the authors investigate the effectiveness of LIPUS and autologous bone grafts in facilitating the healing process of critical size bone defects (CSBD) in rat calvaria. The study posits LIPUS as a potential substitute for autologous bone grafts in bone repair. However, there are concerns regarding certain results and data analysis. A major revision is recommended.
1. In figure 1, panels c and f display inconsistencies in observation magnitude compared to the other panels. It is essential to ensure uniformity across all figures. Please include the full defect along with the surrounding calvaria bone to provide a complete view of the affected area.
2. Figure 2. Please present the BV/TV data with mean and standard deviation
3. Similar issues are noted in Figure 3. To maintain consistency, all figures should be presented at the same observation magnitude. Including full-size images with zoomed-in sections (regional magnifications) would benefit the clarity and detail of the findings.
4. The mechanism of regeneration attributed to intramembranous ossification raises questions. The authors should clarify whether the observed regeneration is a result of the defect model or if LIPUS directly drive intramembranous ossification-based regeneration.
Reviewer 2 Report
Comments and Suggestions for Authors
Comments to the authors
Volaric et al investigated the role of LIPUS for skull bone regeneration using CSBS rat model. It is quite important finding, because, as the authors described, relatively it has not been tested the efficiency of LIPUS during intramembranous ossification. In this paper, the authors compared the regeneration efficiency of LUPUS with AB treatment in vivo, and they concluded both were similarly effective for skull regeneration. The manuscript was relatively written well. However, several concerns were still remained especially about the quality of their data presentation and lacking scientific rigor. Specific comments are listed below.
Specific comments
1) It is important to show the morphological images with lower magnification. This reviewer strongly suggests adding microCT or X-ray images of all groups at all stages.
2) About Figure1, could the authors show the orientation of the sections? Frontal bone? Parietal bone? Also, did the authors find any different regeneration capacity between frontal bones (neural crests derived) and parietal bones (paraxial mesoderm derived)?
3) The text in Figure1b was too small.
4) About table 1, BV/TV cannot be analyzed using HE images, because it was 2D images. Any volume cannot be quantified with 2D image. Could you revise it? Or could you just simply add microCT images?
5) The scale bar of all figures should be with the length, not the magnifications.
6) About Figure3 and 4, basically it was hard to see each cell, such as osteoblast or osteocyte. Could you add the higher magnification images? Also, it was not clearly described how did the authors distinguish each cell (osteoblast, fibroblast, osteocyte, inflammatory cells) and tissue (blood vessels, fibrous tissue, autologous bone, new bone and lamellar bone)? Did the authors check any markers to identify each cell or tissue?
7) About Figure3d,g and Figure4d,g, how can the authors interpret that the intensity of both COX-2 and OSX in AB group were lower than control group at Day7?
8) The order of table2 should be opposite, COX-2 first and OSX second, based on the order of Figures.
9) Could you change table2 to the graphs? It was hard to understand the meaning of p-value (control vs LIPUS, AB vs LIPUS, or all?).
Round 2
Reviewer 2 Report
Comments and Suggestions for Authors
The authors have addressed all of the comments from this reviewer.